

# Wet/dry status change in global closed basins between the mid-Holocene and the Last Glacial Maximum and its implication for future projection

Xinzhong Zhang[1], Yu Li[1], Wangting Ye[1], Simin Peng[1], Yuxin Zhang[1], Hebin Liu[1], Yichan Li[1], Qin Han[1], Lingmei Xu[1]

[1]Key Laboratory of Western China's Environmental Systems (Ministry of Education), College of Earth and Environmental Sciences, Center for Hydrologic Cycle and Water Resources in Arid Region, Lanzhou University, Lanzhou 730000, China

*Correspondence to*: Yu Li (liyu@lzu.edu.cn)

**Abstract.** Closed basins, mainly located in subtropic and temperate drylands, have experienced alarming decline in water storage in recent years. However, a long-term assessment of hydroclimate changes in the region remains unquantified at a global scale. By intergrating the lake records, PMIP3/CMIP5 simulations and modern observations, we assess the wet/dry status during the Last Glacial Maximum, mid-Holocene, pre-industrial, 20th and 21th century periods in global closed basins. Results show comparable wetting at a global scale during the mid-Holocene and modern warming periods with regional mechanism differences, attributed to the boreal winter and summer precipitation increasing, respectively. The long-term moisture change pattern is mainly controlled by the millennial-scale insolation variation, which lead to the poleward moving of westerlies and strengthening of monsoons during the interglacial period. However, modern moisture change trends are significantly associated with ENSO in most of closed basins, indicating strong connection with ocean oscillation. Our research suggests that moisture changes in global closed basins are more resilient than previous thought to warm periods.

## 1 Introduction

A great number of observations in last 100 years show that the Earth's climate is now experiencing significant change characterized by global warming (Hansen et al., 2010; Trenberth et al., 2013; Dai et al., 2015; Huang et al., 2016; Li et al., 2018), which is unequivocally induced by the increase in concentrations of greenhouse gases according to the Fifth Assessment Report of the Intergovernmental Panel on Climate Change (IPCC, 2013). Recent studies have indicated increasing drought and accelerated dryland expansion under modern global warming resulting from a higher vapour pressure deficit and evaporative demand (Dai, 2013; Feng and Fu, 2013; Huang et al., 2017). Assessing the impacts of global warming especially on the terrestrial moisture balance is not only one of the most important social and environmental issues but also the basis of future climate projections.

According to Held's hypothesis, rising atmospheric humidity will cause the existing patterns of atmospheric moisture divergence and convergence to intensify, thereby making effective precipitation more negative in the drylands and more



positive in the tropics, now referred to as the wet-gets-wetter, dry-gets-drier (WWDD) paradigm (Held & Soden, 2006). However, this mechanism may be more complex regionally, especially over terrestrial environments, where wet/dry pattern changes over the past decades and in future projections do not follow the proposed intensification trend (Greve et al. 2014; Roderick et al. 2014). To accurately project future terrestrial hydroclimatic changes, past climates may aid in understanding the regional nuances of the WWDD effect (Lowry & Morrill, 2019). Paleolake studies are among the first to clearly recognize

a fundamental dichotomy in the chronology of high and low latitude moisture balance during the late Quaternary (Street & Grove, 1979), which show high lake levels during glacial maximum at high latitudes, and during interglacials or interstadials in the low-latitude tropics. Then, Quade and Broecker (2009) test Held's hypothesis by taking the Last Glacial Maximum (LGM) as a reverse analog for modern global warming, and point out that the hydroclimate changes in subtropical regions are more complicated. Besides, the African Humid Period and a following mid-Holocene thermal maximum are also the focused

key periods (Lézine et al., 2011), indicating that gradual climate forcing can result in rapid climate responses and a remarkable transformation of the hydrologic cycle (deMenocal & Tierney, 2012). Furthermore, Burke et al. (2018) compared the six warm periods in the past, the early Eocene, mid-Pliocene, Last Interglacial, mid-Holocene (MH), pre-Industrial (PI) and 20th century with the warm periods in the future scenario model to discuss that which provides the best analogs for near-future climates. Previous studies focus more on specific periods and regions, thus a long-term and large-scale evaluation on global hydroclimate

change is of vital significance for the comprehensive understanding the impact of global warming.

       Closed basins account for about one fifth of the global land areas and are mainly located in the arid and semi-arid climate zones. The hydrological cycle of the closed basins is sensitive to climate change, however, their ecosystems play an important role in mitigating global changes by influencing the trend and interannual variability of the terrestrial carbon sink (Ahlström et al., 2015; Li et al., 2017). As there is no outlet or hydrological connection to the oceans, the terminal lakes function as the

ocean for closed basins and concentrate the sedimentary information of the whole basin (Li et al., 2015), which makes them ideal candidates for studying the hydroclimate change of the past. From this perspective, the Earth's surface can be divided into only two parts, the endorheic and exorheic basins, affecting each other through exchanging mass and energy all the time. If it is getting wetter/drier which could result in more/less water stored in endorheic basins, that means more/less water is losing from exorheic basins (oceans). In the most recent IPCC sea level budgets, changes in terrestrial water storage driven by

the climate have been assumed to be too small to be included (IPCC, 2013; Zhan etal., 2019). However, recent advances in gravity satellite measurement enabled a quantification that water storages in closed basin are declining at alarming rates, which not only exacerbate local water stress, but also impose excess water on exorheic basins, leading to a potential sea level rise that matches the contribution of nearly half of the land glacier retreat (excluding Greenland and Antarctica) (Wurtsbaugh et al., 2017; Wang et al., 2018). The impact of global warming on water availability in closed basins is far more serious than that

in other regions, and understanding its hydroclimate change pattern and mechanism differences in the past and modern warm periods will be the key to assess the impact of future climate change.

       In this paper, we focus on the wet/dry pattern changes during the mid-Holocene and modern warming periods from global closed basins to improve our understanding of its response to global warming. Based on the lake records, modern observations



and simulations of the key periods from the Paleoclimate Modeling Intercomparison Project Phase 3 (PMIP3) and Coupled

Model Intercomparison Project Phase 5 (CMIP5), an asessment of hydroclimate change at different timescales from the LGM to MH and PI to late 21st century is conducted. The possible linkages of these moisture change patterns and their underlying physical mechanisms are also discussed. This assessment is essential for future climate projection and regional water management, especially in the dry hinterland.

## 2 Data and methods

### 2.1 Water level and moisture change inferred from lake records

Closed basin was characterized by the collection of sediments and water in the terminal region, thus, the moisture balance of the basin could be primely reflected by the terminal lake records. The following criteria were used for the selection of the proxy records in this study: (1) The proxies should be indicative of moisture changes. (2) The records should cover both the LGM and MH time slices. (3) The dominant driving mechanism of the variation in proxy records should be climatic changes.

(4) The records should have a dating control level of 6 or better according to the Cooperative Holocene Mapping (COHMAP) project dating scheme. A dating control level of 6 applied to continuous sequences such as sediment cores corresponds to bracketing dates, generally one within 6000 years and the second within 8000 years of the time being assessed; the same control level applied to discontinuous sequences, such as dated shorelines, requires at least one date within 2000 years of the time being assessed (Lowry & Morrill, 2019). Finally, 52 moisture change records from the recently published literatures

(Table 1) and 50 water level records from the Global Lake Status Data Base (Kohfeld and Harrison, 2000; Harrison et al., 2003) and Chinese lake-status database (Yu et al., 2001; Xue et al., 2017) in global closed basins and surrounding areas were selected (Fig 1). To capture the general spatial pattern, the differences of lake status between the LGM and MH in individual records were classified into 3 grades (higher/wetter, moderate, lower/drier) and compared with the modeled direction of effective precipitation changes from PMIP3/CMIP5 simulations.


**Table 1**. Moisture change records from the recently published literatures. "+"/ "-" indicates wetter/drier climate condition from records or more/less effective precipitation from multi-models during the MH than that during the LGM.

| Lake name | Lon (°E) | Lat (°N) | Elev (m) | Records | Models | References |
|-----------|----------|----------|----------|---------|--------|------------|
| Surprise | -120.1 | 41.5 | 1370 | - | - | (Ibarra et al., 2014) |
| Lahontan | -119.5 | 40 | 1180 | - | - | (Benson et al., 2013) |
| Owens | -119 | 38 | 1080 | - | - | (Bacon et al., 2003) |
| Mojave | -116.8 | 36 | -60 | - | - | (Wells et al., 2003) |
| Franklin | -115.3 | 40.3 | 1820 | - | - | (Munroe & Laabs, 2013) |
| Clover | -114.6 | 40.9 | 1700 | - | - | (Munroe & Laabs, 2013) |
| Bonneville | -113 | 40.5 | 1280 | - | - | (Oviatt, 2015) |
| Estancia | -105.6 | 34.6 | 1860 | - | - | (Allen & Anderson, 2000) |





| | | | | | |
|---|---|---|---|---|---|
| Santiaguillo | -104.8 | 24.8 | 1960 | - | - | (Chávez-Lara et al., 2015) |
| Pátzcuaro | -101.6 | 19.6 | 2040 | - | + | (Bradbury, 2000) |
| Huelmo | -73 | -41.5 | 10 | - | - | (Massaferro et al., 2009) |
| Tagua Tagua | -71.2 | -34.5 | 200 | - | - | (Valero-Garcés et al., 2005) |
| Potrok Aike | -70.4 | -52 | 110 | - | - | (Kliem et al., 2013) |
| Cari Laufquen | -69.6 | -41.4 | 790 | - | + | (Cartwright et al., 2011) |
| Titicaca | -69.4 | -16 | 3800 | - | + | (Rowe et al., 2002) |
| Uyuni | -67.5 | -20.2 | 3650 | - | - | (Baker et al., 2001) |
| Pozuelos | -66 | -22.4 | 3660 | - | + | (McGlue et al., 2013) |
| Bosumtwi | -1.4 | 6.5 | 150 | + | + | (Shanahan et al., 2006) |
| Chad | 14 | 13 | 280 | + | - | (Armitage et al., 2015) |
| Ngami | 22.7 | -20.5 | 920 | - | - | (Burrough et al., 2007) |
| Tanganyika | 29.8 | -6.7 | 773 | + | + | (Felton et al., 2007) |
| Albert | 31 | 1.5 | 615 | + | + | (Talbot et al., 2000) |
| Rukwa | 32 | -8 | 800 | + | + | (Thevenon et al., 2002) |
| Victoria | 33 | -1 | 1135 | + | - | (Talbot & Lærdal, 2000) |
| Tuz | 33.4 | 38.7 | 905 | - | - | (Doğan, 2010) |
| Masoko | 33.8 | -9.3 | 840 | + | + | (Garcin et al., 2006) |
| Malawi | 34.23 | -10 | 468 | + | - | (Johnson et al., 2002) |
| Lisan | 35.5 | 31.5 | -430 | - | - | (Bartov et al., 2002) |
| Turkana | 36.1 | 3.6 | 360 | + | + | (Morrissey et al., 2014) |
| Challa | 37.7 | -3.3 | 880 | + | - | (Moernaut et al., 2010) |
| Abiyata | 38.7 | 7.7 | 1573 | + | + | (Chalié & Gasse, 2002) |
| Van | 43 | 38.5 | 1640 | + | - | (Çağatay et al., 2014) |
| Urmia | 45.5 | 37.5 | 1267 | - | + | (Stevens et al., 2012) |
| Zeribar | 46 | 35.5 | 1285 | - | - | (Stevens et al., 2001) |
| Caspian Sea | 50.7 | 41.7 | -28 | + | - | (Yanina et al., 2014) |
| Aral Sea | 60 | 45 | 42 | + | - | (Boomer et al., 2000) |
| Karakul | 73.5 | 39 | 3915 | + | - | (Heinecke et al., 2017) |
| Son Kul | 75 | 41.8 | 3016 | + | + | (Huang et al., 2014) |
| Issyk-Kul | 77.3 | 42.4 | 1607 | + | + | (Ricketts et al., 2001) |
| Zabuye | 84 | 31.6 | 4421 | - | + | (Wang et al., 2002) |
| Bosten | 87 | 42 | 1048 | + | - | (Huang et al., 2009) |
| Nam Co | 90.5 | 30.7 | 4718 | - | + | (Mügler et al., 2010) |
| Lop Nur | 91 | 40.8 | 780 | - | + | (Chao et al., 2009) |
| Hurleg | 96.9 | 37.3 | 2817 | + | - | (Zhao et al., 2007) |
| Chaka | 99.1 | 36.7 | 3200 | - | + | (Liu et al., 2008) |
| Genggahai | 100 | 36.1 | 2860 | - | + | (Qiang et al., 2013) |
| Qinghai | 100 | 38 | 3260 | + | + | (Jin et al., 2015) |
| Khubsugul | 100.5 | 51 | 1645 | + | + | (Fedotov et al., 2004) |
| Juyanze | 101.5 | 41.8 | 900 | - | - | (Hartmann et al., 2009) |





| Eyre | 137.4 | -28.4 | -15 | - | - | (Magee et al., 2004) |
| Frome | 139.9 | -30.6 | 1 | - | - | (Deckker et al., 2011) |
| Callabonna | 140 | -29.7 | 1 | - | - | (Cohen et al., 2012) |

## 2.2 Modern data sources and analyses

Closed basin extents were acquired from HydroBASINS product, a series of polygon layers that depict watershed boundaries and sub-basin delineations at a global scale by using the HydroSHEDS (Hydrological data and maps based on SHuttle Elevation Derivatives at multiple Scales) database at 15 arc-second resolution (Lehner et al., 2013). There were some exceptions we did not take them into account in this study: (1) Ten landlocked watersheds in the Inner Tibetan Plateau, Northeast China, Siberia and western United States were captured only in Global Drainage Basin Database (Masutomi et al., 2009; Wang et al., 2018); (2) Sporadic landlocked watersheds smaller than 100 km$^2$ Embedded in the exorheic regions were not considered as independent units; (3) Some of the contemporary endorheic watersheds were exorheic in the past, such as the Wuyuer river basin in Northeast China.

Primary variables of mean precipitation (P) and potential evapotranspiration (PET) from Climatic Research Unit Time-Series version 4.01(CRU TS4.01), a gridded time-series dataset of month-by-month variation in climate covering all land areas (excluding Antarctica) at 0.5° resolution over the period 1901-2016 (Harris et al., 2014), were used for modern climate analysis. Aridity index (AI) defined as the ratio of annual precipitation to annual potential evapotranspiration by the United Nations Environment Programme (UNEP, 1992) were applied. Furthermore, to explore the possible relationship between the ocean and closed basins in modern times, we conducted pearson correlation analysis between monthly AI value and multivariate El Niño/Southern Oscillation (ENSO) index (MEI) (Kobayashi et al., 2015) for different endorheic regions during 1979-2016. Linear trend analysis was used, and a trend was considered statistically significant at a significance level of 5%.

## 2.3 Debiasing and downscaling of PMIP3/CMIP5 multi-model ensemble

Experiments of the LGM, MH and PI from the PMIP3 and projection experiment of 21st century under Representative Concentration Pathway 8.5 (RCP8.5) from the CMIP5 were involved in this study (Braconnot et al., 2012; Taylor et al., 2012). To ensure the consistency and precision of simulations, we used the outputs from 5 global climate models (Table 2) which have all completed the above key period experiments at a spatial resolution of less than 2 degrees. The periods of 2006-2015 and 2091-2100 were defined as the representatives of early and late 21 century (E21 and L21), respectively.

Statistical downscaling and debiasing followed a multi-step approach described by Tabor and Williams (2010). The primary climate variables were first debiased by differencing each paleoclimate (LGM, MH, PI) or future climate (2017-2100) simulation from a present climate simulation (2006-2015). These anomalies are then downscaled through spline interpolation to a 0.5° resolution grid corresponding to the modern observational CRU dataset. The anomalies are then added to the observational data (2006-2015) to produce the debiased and downscaled primary variables for the paleoclimate or future climate simulation. This differencing removes any systematic difference as long as that bias is constant through time (Wilby





et al,. 2004; Lorenz et al., 2016). The effective precipitation calculated by precipitation minus evaporation was introduced to compare with the lake status during the LGM and MH, and predict future changes in moisture balance.

**Table 2.** PMIP3/CMIP5 models used in this study.

| Model name | Resolutions | Modelling centre |
|---|---|---|
| CCSM4 | 288×192×L26 | National Center for Atmospheric Research, USA |
| CNRM-CM5 | 256×128×L31 | Centre National de Recherches Meteorologiques, France |
| GISS-E2-R | 144×90×L40 | NASA Goddard Institute for Space Studies, USA |
| MIROC-ESM | 128×64×L80 | Japan Agency for Marine-Earth Science and Technology, Japan |
| MRI-CGCM3 | 320×160×L48 | Meteorological Research Institute, Japan |

## 3 Results

### 3.1 Evidence from lake records

Generally, lake level changes match climate changes from the proxy record well (Fig 1), except for Western China where plenty of hydroclimate records from different lakes remain controversial. In the North and South American continents, almost
all closed basins have experienced a wetter LGM compared to MH, the same as the Eastern Mediterranean region and south Tibetan Plateau. On the contrary, Eastern African highlands and the Sahel region show a prevailing wetter MH, which is highly attributed to the African Humid Period. The monsoonal Eastern Asia and arid Central Asia both record a wetter MH in the lake level and basin climate change, respectively. In the middle area between the above regions, there are some contradictory records synchronously show lower lake level and wetter climate conditions. Records from Southern Africa and Australia are
insufficiency, and these several evidences tend to support a wetter LGM. Thus, there will be two noticeable latitude belts at around 30 degrees where substantial high lake levels during the LGM have disappeared or subsided during the MH. However, the low-latitude Africa and mid-latitude Asia show the opposite pattern which experienced a wetter MH.

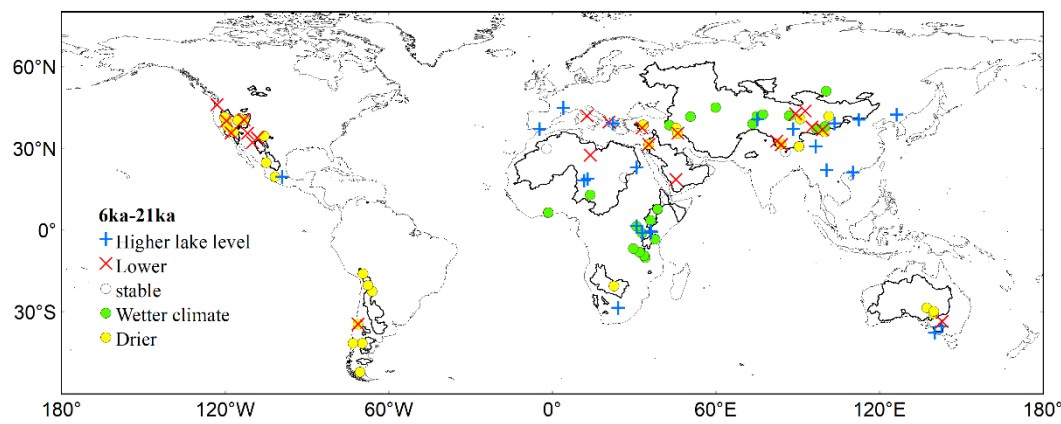





**Figure 1.** Wet/dry status change between the LGM and MH from lake records. The blue and red cross sites are from lake status
databases; the green and yellow point sites are from recently published literatures; the hollow points indicate that there is no
significant change in lake level or climate condition between the LGM and MH.

We compare the modeled direction of effective precipitation change to lake records from 52 sites we compiled from the
published literature, and the model ensemble does particularly well in simulating the direction of hydroclimate change in most
of closed basins. One important mismatch of the model ensemble with the lake record occurs in the Central Eurasia, as the
lake record suggests wetter climate condition at MH, contrary to the model ensemble. Excluding that parts, the consistency
between models and records will be more than 80%. Some minor mismatches occur in the East Africa and South America,
where the altitude changes dramatically so that the models appear to miss the details of climate change.

### 3.2 Climate changes under the past and modern warming from multi-models

We assess the annual mean temperature, precipitation and effective precipitation changes of total closed basins under the two
global warming processes, results show increases of 3.2℃, 97.8 mm, 11.5 mm from LGM to MH and 4.3℃, 45.0 mm, 10.0
mm from PI to L21. In the other word, the humans will spend less than three hundred years to make temperature of global
closed basins rise more than the past twenty thousand years under RCP8.5 scenario. From LGM to MH, annual precipitation
increases twice than that from PI to L21, but annual effective precipitation changes of the two periods are almost close,
indicating the evaporation factor may play a dominant role.

As shown in Fig 2, there are some similar patterns of hydroclimate change between MH-LGM and L21-PI, however, some
notable spatial and temporal differences still exist. The MH warming is characterized by strong latitudinal zonality, resulting
in the most strong warming at the high-latitudes in the North Hemisphere. While the modern warming from PI to L21 is more
homogeneous over all the closed basins. The patterns of precipitation change are similar between the two warming periods,
except for the belt region from Mexican Plateau to the Mediterranean region and Iranian Plateau where precipitation is
increasing from PI to L21. The strong increasing in Central Asia and decreasing in Southern Africa are shown during the both
periods. It is more complex for the effective precipitation changes. One important difference occurs in the Central Asia, where
strong drought trend prevails from PI to L21, contrary to the MH warming. Moreover, drought in the Western America,
Southern Africa and Australia will be lower in the future warming.

Monthly precipitation and evaporation changes of the total closed basins in different warming periods are shown in Table
3. There is an apparent seasonal difference that precipitation and evaporation increasing mainly happen in the boreal summer
half-year during the MH warming, while in modern and future warming periods they concentrate on the boreal winter half-
year. In this century, it seems to be more even in all seasons. Besides, the only month of precipitation decreasing occurs in
March during the MH warming, but in June during modern and future warming. In the MH and future warming, the amplitude
of evaporation increasing is always greater than that of precipitation increasing, indicating more drought stress.





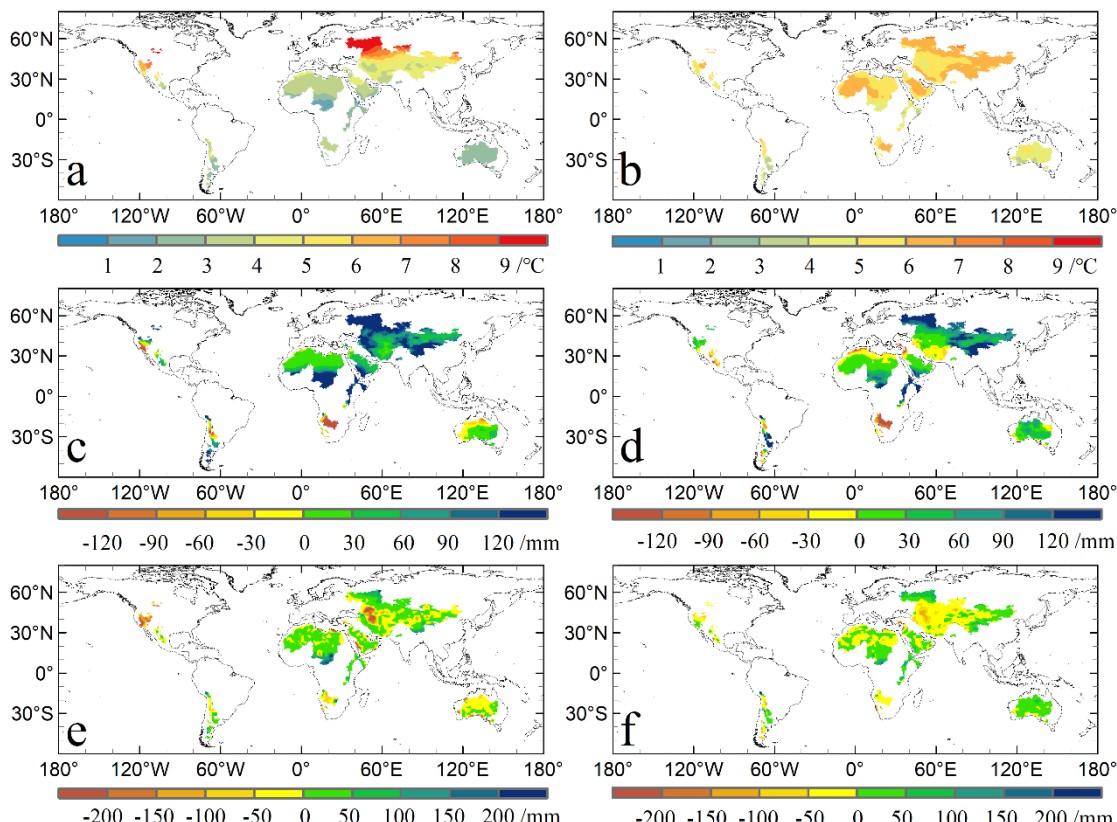

**Figure 2.** Annual mean temperature (a,b), precipitation (c,d) and effective precipitation (e,f) differences for MH-LGM (left) and L21-PI (right) from the PMIP3/CMIP5 multi-model ensemble.

**Table 3.** Percentage changes of monthly precipitation and evaporation between different periods from the multi-models.

|  |  | 1 | 2 | 3 | 4 | 5 | 6 | 7 | 8 | 9 | 10 | 11 | 12 |
|---|---|---|---|---|---|---|---|---|---|---|---|---|---|
| MH-LGM | P | 8.3 | 2.0 | -1.1 | 9.7 | 25.8 | 32.1 | 41.1 | 52.7 | 57.5 | 46.0 | 34.8 | 18.1 |
|  | E | 17.1 | 11.0 | 10.9 | 16.0 | 27.4 | 36.1 | 40.2 | 43.7 | 42.4 | 35.9 | 30.7 | 24.9 |
| L21-PI | P | 10.8 | 12.1 | 12.6 | 14.4 | 8.3 | -1.5 | 1.5 | 6.5 | 8.3 | 11.3 | 11.6 | 11.4 |
|  | E | 9.5 | 10.2 | 11.6 | 14.5 | 12.8 | 9.5 | 6.5 | 7.9 | 6.8 | 4.9 | 6.5 | 9.5 |
| L21-E21 | P | 9.3 | 7.7 | 8.9 | 12.7 | 7.3 | -1.0 | 4.6 | 8.2 | 6.2 | 8.0 | 6.5 | 7.9 |
|  | E | 8.4 | 7.9 | 8.3 | 12.0 | 10.7 | 8.8 | 6.7 | 8.4 | 6.7 | 5.4 | 4.6 | 7.1 |

**3.3 Recent moisture trends**

In the 20th Century, the annual mean AI of global closed basins slowly increases at a rate of 0.01 every 100 year, generally showing a positive response to the global warming as it is from the LGM to MH and from PI to L21 . At the more recent decadal timescale during 1979-2016, as shown in Fig 3, most of closed basins in Americas and Central Eurasia are experiencing

the worst drought trend, while the low-latitudes of Americas and the third pole region including Tianshan Mountains, the





Pamir and Tibetan Plateau show a wetting pattern at the same time. Also, the Sahel, Horn of Africa, Southern Africa and Northern Australia are getting wetter during the past several decades. The important mismatch with the hydroclimate change pattern under the past and future warming occurs in the Southern Africa, as it is getting wetter contrary to the latters. It is worth noting that only trends in the Northern Africa, Arabian Peninsula and Iranian Plateau reach 0.05 significance level, indicating

that the hydroclimate change in other parts of global closed basins are unstable and uncertain.

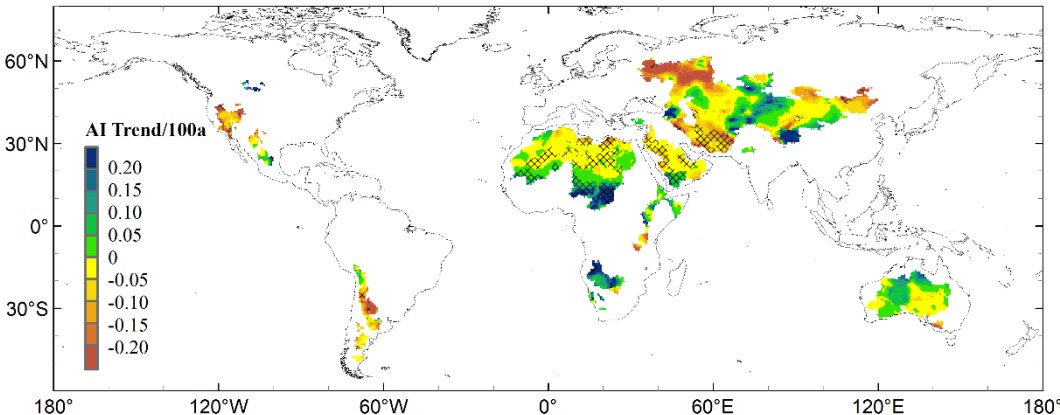

**Figure 3.** Modern observational aridity index linear trend during 1979-2016 from CRU Dataset. Gridding indicate that the trends are statistically significant at 5% level.

For the whole closed basins, the moisture change is significantly positive related to monthly MEI from August to December, and the highest relation occurs in December, the boreal winter season. The Australian closed basins are lightly negative correlated with monthly MEI from September to December, the local autumn time. It is well known that during an ENSO warm event, drought occurs in regions of northeastern Australia, leading to anomalously low annual rainfall (Cai et al., 2001; King et al., 2014). Thus the precipitation or runoff from the upstreams in the north mainly control the moisture fluctuations in

the Australian closed basins. While the positive relationship between AI from the Central Eurasia and MEI is keeping in half-year period from June to December, with the highest relation occurs in July. The ENSO-based composite analyses have shown that these water vapor fluxes of these seasonal precipitation are mainly generated in Indian and North Atlantic Oceans and transported by enhanced westerlies during EI Nino (Xi et al., 2018; Rana et al., 2017). The most strong correlations exist in the closed basins of North America and Southern Africa, which are significantly positive and negative related to the ENSO,

with both the highest correlation in boreal spring season (Holmgren et al., 2006; Cook et al., 2000). The rest parts of global closed basins show no significant correlation with any monthly EMI.

**Table 4.** Pearson correlation coefficients between endorheic basin AI and monthly MEI during 1979-2016. The bold numbers mean that correlation coefficients are statistically significant at 5% level. SAM-South America, NAM-North America, SAF-



Southern Africa, EAF-Eastern Africa, NAF-Northern Africa and Arabian peninsula, CEA-Central Eurasia, AUS-Australia, ALL-Global endorheic basins.

| Month | 1 | 2 | 3 | 4 | 5 | 6 | 7 | 8 | 9 | 10 | 11 | 12 |
|---|---|---|---|---|---|---|---|---|---|---|---|---|
| SAM | -0.23 | -0.26 | -0.15 | -0.05 | -0.05 | 0.08 | 0.22 | 0.29 | 0.27 | 0.22 | 0.14 | 0.19 |
| NAM | **0.47** | **0.52** | **0.55** | **0.50** | **0.47** | 0.30 | 0.06 | -0.02 | 0.01 | -0.01 | -0.01 | -0.02 |
| SAF | **-0.60** | **-0.57** | **-0.60** | **-0.64** | **-0.53** | **-0.37** | -0.15 | 0.01 | -0.07 | -0.10 | 0.00 | 0.00 |
| EAF | -0.02 | -0.06 | -0.12 | -0.07 | 0.01 | -0.01 | -0.02 | 0.08 | 0.10 | 0.16 | 0.25 | 0.28 |
| NAF | 0.03 | 0.01 | -0.03 | -0.09 | -0.09 | -0.05 | 0.00 | 0.01 | 0.05 | 0.10 | 0.14 | 0.11 |
| CEA | 0.19 | 0.16 | 0.13 | 0.19 | 0.31 | **0.45** | **0.50** | **0.47** | **0.47** | **0.45** | **0.43** | **0.41** |
| AUS | 0.06 | 0.09 | 0.09 | 0.13 | 0.06 | -0.13 | -0.28 | -0.28 | **-0.32** | **-0.34** | **-0.37** | **-0.36** |
| ALL | -0.02 | -0.05 | -0.09 | 0.02 | 0.13 | 0.24 | 0.30 | **0.38** | **0.36** | **0.37** | **0.41** | **0.42** |

## 4 Discussion

Recognition that Earth orbital changes are the basic cause for Quaternary climatic variations provides a context for explaining global environmental changes, many of which are preserved in the stratigraphic and geomorphic records of lakes (Wright,
1996). As mentioned before, asynchronous warming whithin land or between land and sea can results in changes in the pattern of regional atmospheric circulation. Closed basins are mainly located in subtropic and temperate drylands, where the interactions of westerly and monsoon system are strong. The positions and intensity of them will determine the local moisture balance patterns in most of closed basins.

Evidences from the paleoclimate and archaeological records in Northern Africa had shown that the world's largest desert in
modern times was covered by numerous forests and lakes paced by earth's orbital changes during the early Holocene (deMenocal & Tierney, 2012), and this kind of changes were reflected in most subtropical monsoon regions all over the world. As the most typical one, Asian monsoon even reached as far as the west Tianshan mountains during the early and middle Holocene (Wang et al., 2014), bringing monsoon precipitation and affecting the wet/dry patterns in the eastern region of central Eurasia. Australian closed basins were significantly affected by the subtropical Australian monsoon as well, due to the main
river flow direction that was from the north to south, and the water level of Lake Eyre in the early and middle Holocene reached the highest during the past 30 thousand years (Magee et al., 2004). Similarly, high lake levels and a wetter climate turned back in the MH over North American monsoon region, however, some studies suggested that this trend is caused by the destruction of vegetation by early human activities (Bridgwater et al., 1999; Caballero et al., 1999).

A strong hemispheric symmetry of drought in MH from lake records exists at about 30° latitudes, and the belt in the South
Hemisphere is closer to the equator than that in the North Hemisphere. The relatively higher lake level or wetter climate condition during the LGM is likely attributed to the equatorward moving of the westerlies. This pattern is more significant in North America because of the existence of Laurentian Ice Sheet (Lowry & Morrill, 2019). Positive hydroclimate change in the Northern and Eastern Africa have already verified by the studies on the strengthening of the West African monsoon during the MH (Lézine et al., 2011; deMenocal & Tierney, 2012). In addition, it was also considered that the increase of atmospheric



$CO_2$ concentration was the main driving force for the changes of climate and lake level since the LGM (Shakun et al., 2010;
        Li et al., 2013).

        However, changes in the Central Euraisa are complicated due to the interactions of westerly and monsoon in the middle
        region between the arid Central Asia and monsoonal Eastern Asia. They both get wetter during the MH because of the Central
        Asia westerly winds moving and the East Asian monsoon strengthening. The transition areas showed some drought trends
which can be explained by less monsoon precipitation during MH and more westerly winds precipitation during LGM in this
        region. During the LGM period, the westerlies in the Northern Hemisphere moved south to the southwest of the United States
        and the eastern Mediterranean region (Lachniet et al., 2014; Claire et al., 2010). The effective precipitation in the arid Central
        Asia didn't increase until the middle and late Holocene, leading to a large number of low lake levels. However, due to the
        increase of summer solar radiation in the northern hemisphere during the early and middle Holocene, a stronger East Asian
monsoon brought more precipitation and high lake levels, showing different climate response patterns compared with that in
        the arid Central Asia (Chen et al., 2008; Ran et al., 2013; Huang et al., 2014).

        In the last century, the Americas, Central Eruasia and Australia have experienced a significant wetting trend, and resulting
        in a wetting in closed basins globally. The North Africa and East Asia where are generally influenced by strong summer
        monsoons are no longer wetter from the observations. However, from the AI projection during 207-2100, it is worth noting
that the Mexican Plateau, Sahel, Horn of Africa in the North Hemisphere and the Altiplano, Southern Africa and Northern
        Australia in the South Hemisphere seems to form a new hemispheric symmetry in subtropics. In modern warming period from
        1901 to 2100, the winter precipitation play a dominant role in determining the wet/dry pattern change in closed basins, implying
        the significance of westerly instead of monsoon.

        As the hydroclimate change patterns in different latitudes at the millennial, centurial and decadal timescales have shown
considerable connection with the general atmospheric circulation (Tierney et al., 2013; Ljungqvist et al., 2016; Zhang et al.,
        2017; Kohfeld et al., 2013), such as the westerly winds and monsoons, we have tested the relationship between AI and various
        marine and terrestrial climate index. However, the only significant connection with ENSO is shown in Table S3. It is well
        known that during an ENSO warm event, drought occurs in regions of northeastern Australia, leading to anomalously low
        annual rainfall (Cai et al., 2001; King et al., 2014). Thus the precipitation or runoff from the upstreams in the north mainly
control the moisture fluctuations in the Australian endorheic basins. The ENSO-based composite analyses have shown that the
        water vapor fluxes of seasonal precipitation in Central Eurasia are mainly generated in Indian and North Atlantic Oceans and
        transported by enhanced westerlies during EI Nino (Xi et al., 2018; Rana et al., 2017). However, the rest parts of closed basins
        show no significant correlation with any monthly EMI. These patterns provide some new perspectives to understand the
        differences and connections over global endorheic basins.



## 5 Conclusion

In summary, this study presents a comprehensive analysis of hydroclimate change at different timescales in global closed basins. The patterns of hydroclimate changes during the mid-Holocene and modern warming periods show comparable spatio-temporal consistency. But on the seasonal characteristic, the precipitation increasing concentrates in the boreal winter from LGM to MH, on the contrary to the modern warming period. The seasonal difference of precipitation increasing may also indicate the different dominant roles of westerly winds and monsoons during the two periods. Our results suggest that the long-term regional differences of hydroclimate change are mainly controlled by the millennial insolation variation, which leads to equatorward moving of the westerlies during the glacial period and the strengthening of monsoons during the interglacial period. While during the modern warming period, regional differences of moisture change are more localized, and most of closed basins show connections with ENSO. We conclude that moisture changes in global closed basins are more resilient than previous thought to global warming and more integrated studies are necessary for the future projection.

*Data Availability.* Global closed basins boundaries are available from the Hydrological data and maps based on SHuttle Elevation Derivatives at multiple Scales (HydroSHEDS) website https://www.hydrosheds.org/page/hydrobasins. The Global Lake Status Data Base and Chinese lake-status database are available from the Paleoclimatology Datasets of NOAA's National Centers for Environmental Information (NCEI) https://www.ncdc.noaa.gov/data-access/paleoclimatology-data/datasets. CRU TS4.01 data are available from https://crudata.uea.ac.uk/cru/data/hrg/. MEI.v2 Values are available from https://www.esrl.noaa.gov/psd/enso/mei/. PMIP3/CMIP5 simulations are available from the Earth System Grid Federation (ESGF) Peer-to-Peer (P2P) enterprise system website https://esgf-node.llnl.gov/projects/esgf-llnl/.

*Author contributions.* Yu Li and Xinzhong Zhang designed this study and carried it out. Wangting Ye, Yuxin Zhang and Simin Peng contributed to the data processing, analysis and discussion of results. Xinzhong Zhang prepared the manuscript with contributions from all co-authors.

*Competing interests.* The authors declare that they have no conflict of interest.

*Acknowledgements.* This work was supported by the National Natural Science Foundation of China (Grant Nos. 41822708 and 41571178), the Strategic Priority Research Program of Chinese Academy of Sciences (Grant No. XDA20100102), the Fundamental Research Funds for the Central Universities (Grant No. lzujbky-2018-k15) , and the Second Tibetan Plateau Scientific Expedition (STEP) program (Grant No. XDA20060700).



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
