# Peer review of "Wet/dry status change in global closed basins between the mid-Holocene and the Last Glacial Maximum and its implication for future projection"

_Climate of the Past, 2020_

## Referee Comment (RC1) · Cody Routson (Referee) · 20 Mar 2020

Zhang et al., present a nice new compilation of existing Holocene and glacial hydroclimate records, which is accompanied by an interesting analysis. They compare proxy hydroclimate records, proxy lake level records, and PMIP simulations between the last glacial maximum, mid-Holocene, and future warming. Generally the analyses are straight forward and I think worthwhile of publication. However, before I can adequately evaluate the study design and associated conclusions, much work needs to be done to clarify the writing. The manuscript would substantially improve by having a native english speaker edit the sentences and overall structure. Many of the sentences

are incomplete, difficult to follow, or entirely nonsense. I would also highly recommend reading and implementing the writing principals outlined by Joshua Schimel in his book "Writing Science" published in 2012.

Below is my preliminary review, which was conducted rapidly and remains incomplete until the overall presentation and writing is improved.

The introduction is meandering and hard to follow. Make sure each sentence doing work, frame the knowledge gap, and keep the story moving forward in a logical sequence.

Line 62. This gets to the point of the study, but should also include the LGM. Something along the lines of "…pattern of changes during the LGM, mid-Holocene and modern warm period…"

Also there is only one modern warm period so it should be singular in this sentence.

Line 71: This sentence is difficult for me to follow. Please re-write. Or remove?

Line 72: Is there a sampling resolution criteria?

Line 76: I appreciate a description of the COHMAP dating scheme, however it is difficult to follow as written. Please clarify. Use multiple sentences if needed.

Line 79. It took me some time to figure out what what you are trying to communicate here. The finding of 52 sites and Table 1 are results, and should be moved down into the beginning of the results section. Then, on line 79, a new paragraph should be started with rewording the sentence to something along the lines of: "We then compared our new compilation of proxy records to 50…"

Line 82-84: Sentence structure, please clarify. Use multiple sentences as necessary.

Line 107: Replace "involved" with "used"

Line 127: This sentence doesn't make sense.

Line 148: Please show the data (a graph or otherwise) to support the statement in this paragraph.

Line 145: I'm not following the argument in this paragraph. What two global warming processes? Was this described somewhere in the methods? E21 and L21? Please clarify.

Line 51 and Figure 2: Please justify the comparison between warming from the LGM to MH versus PI to future warming. This should be done in the methods and then discussed in the discussion. There are very important mechanistic differences between mid-Holocene and future warming. Mid-Holocene warming was driven by changes in primarily summertime insolation whereas future warming is driven by greenhouse forcing. Some impacts are comparable, but differences in forcing mechanisms need addressed. It appears that you try to do some of this in the discussion, but it needs to be developed/clarified.

Figure 2 and 3 captions. Please indicate why the maps have extreme missing data coverage. The significant regions are shown by the gridding... CRU has data over the regions which show no data...

Line 193: Please point the reader to the correlations (Table 4) before discussing them.

Line 241: Please point toward something to justify the statement that winter precipitation will play a dominant roll in future hydroclimate changes.

Line 256: Please remove the word "comprehensive".

Line 266: The conclusion that moisture changes in closed basins are resilient to warming needs justified... Large increases in temperature alone will dramatically increase evaporation and decrease effective moisture (e.g. lake level) under RCP 8.5 scenarios.

Data availability: Please make your compilation of 52 hydroclimate records available in addition to data that were already available.

In general, make sure statements are accompanied by the data that support them (Figures, tables or otherwise).

I look forward to clarifications and revisions,

Sincerely,

Cody Routson

---

## Referee Comment (RC2) · Anonymous Referee #2 · 24 Apr 2020

Dear editor and authors of the manuscript "Wet/dry status change in global closed basins between the mid-Holocene and the Last Glacial Maximum and its implication for future projection", I have no ability to assess the use of lake sediment data due to my professional restriction, and propose my opinion about the model results. This work is encouraged to help future projection using the paleoclimate information. The Global model-data comparison over the orbital scale and short-scale is important to understand the difference in the local hydroclimate variations. The analysis is logical and fruitful. I was attracted by the idea of the manuscript, but still felt unsatisfied about the mechanism. Thus, I suggest that the manuscript should be accepted for publication after a minor revision.

Main comments:

1. The mechanism should be furtherly improved. e.g. the influence of the insolation is not derived from the current study. A regional difference and its reason should be emphasized in the abstract and the conclusion. e.g. the difference of the hydroclimate variation mechanisms in the Central Eurasia and other regions.

2. The reason for choosing the analysis period is obscure. It is hard to understand to compare a centennial and longer variation of glacial-interglacial cycle with a decadal variation (AD 2006-2015 and AD 2091-2100). Is it better to choose the period as long as possible, e.g. the entire 20th century and 21th century?

3. The conclusion should be furtherly verified. It is challenging that a decadal variation in the late 21th century is attributed to the ENSO variability. The Pacific decadal oscillation or the Inter-decadal pacific oscillation may be more appropriate, if the analyzed period would be extended to the entire century.

Specific Comments:

1. Page 1, line 17. The location of these basins should be provided, which is your contribution. e.g. There is an opposite significant AI-MEI relationship between in the Southern Africa and in the Central Eurasia.

2. Page 2, line 30. The abbreviation of the term 'wet get wetter dry get drier' could be revised to 'DGDWGW' according to the previous study [Hu et al., 2019].

3. Page 2, line 56. 'Zhan et al.,'

4. Page 3, line 73-75. How to know that the proxy should be indicative of moisture changes and its drive is climatic change'? Following the original descriptions?

5. Page 3, line 83. What does the 'direction' mean? Is it a trend?

6. Pages 3-5. The table 1 should be changed with a Figure 1b to show model-data comparison.

7. Page 6, line 120. References of these experiments should be added.

8. Page 7, line 138-143. A new Figure 1b would be helpful to describe this part.

9. Page 7, line 153. The L21-PI difference is a future scenario not a modern warming.

10. Page 8, line 174. Why to select the period 1979-2016? Is it better to choose the same period with the early 21th century (AD 2006-2015)?

11. Page 10, line 205. 'within'

12. Page 11, line 239. '2091-2100'?

13. Page 11, line 242. the period (AD 1901-2100) was not analyzed in this study.

14. Page 11, lines 244-254. If the period is extended to the entire 20th century and 21th century, this discussion may be related to the above mechanism about a centennial and longer variation of glacial-interglacial cycle.

Reference: Hu, Z., X. Chen, D. Chen, J. Li, S. Wang, Q. Zhou, G. Yin, and M. Guo (2019), "Dry gets drier, wet gets wetter": A case study over the arid regions of central Asia, Int. J. Climatol. , 39(2), 1072-1091.

---

## Author Comment (AC1) · 1 May 2020

**Response to Cody Routson**

**We very much thank the reviewer for taking the time to review our manuscript. The comments and suggestions are of tremendous assistance to improve the quality of our manuscript. The point-by-point responses are listed below.**

Zhang et al., present a nice new compilation of existing Holocene and glacial hydroclimate records, which is accompanied by an interesting analysis. They compare proxy hydroclimate records, proxy lake level records, and PMIP simulations between the last glacial maximum, mid-Holocene, and future warming. Generally the analyses are straight forward and I think worthwhile of publication. However, before I can adequately evaluate the study design and associated conclusions, much work needs to be done to clarify the writing. The manuscript would substantially improve by having anative english speaker edit the sentences and overall structure. Many of the sentences are incomplete, difficult to follow, or entirely nonsense. I would also highly recommend reading and implementing the writing principals outlined by Joshua Schimel in his book "Writing Science" published in 2012. Below is my preliminary review, which was conducted rapidly and remains incomplete until the overall presentation and writing is improved.

1. The introduction is meandering and hard to follow. Make sure each sentence doing work, frame the knowledge gap, and keep the story moving forward in a logical sequence.

We revised and streamlined the introduction to improve readability.

2. Line 62. This gets to the point of the study, but should also include the LGM. Something along the lines of "…pattern of changes during the LGM, mid-Holocene and modern warm period…" Also there is only one modern warm period so it should be singular in this sentence.

The sentence was rewritten as suggested.

3. Line 71: This sentence is difficult for me to follow. Please re-write. Or remove?

Removed.

4. Line 72: Is there a sampling resolution criteria?

The dating control level rather than sampling resolution criteria was used here. Given the lack of continuous lake records dating back to the LGM (usually with depositional hiatuses), it's hard to make a resolution limit.

5. Line 76: I appreciate a description of the COHMAP dating scheme, however it is difficult to follow as written. Please clarify. Use multiple sentences if needed.

Clarified using multiple sentences.

6. Line 79. It took me some time to figure out what what you are trying to communicate here. The finding of 52 sites and Table 1 are results, and should be moved down into the beginning of the results section. Then, on line 79, a new paragraph should

be started with rewording the sentence to something along the lines of: "We then compared our new compilation of proxy records to 50…"

A new Figure 1b (model-data comparison based on Table 1) was added, and Table 1 was moved into Supplement. On original line 79, a new paragraph was started with rewording the sentence as suggested.

7. Line 82-84: Sentence structure, please clarify. Use multiple sentences as necessary.

Clarified using multiple sentences.

8. Line 107: Replace "involved" with "used"

Done.

9. Line 127: This sentence doesn't make sense.

The paragraph was reorganized and revised to make it clearer.

10. Line 138: Please show the data (a graph or otherwise) to support the statement in this paragraph.

A new Figure 1b was added.

11. Line 145: I'm not following the argument in this paragraph. What two global warming processes? Was this described somewhere in the methods? E21 and L21? Please clarify.

Two global warming processes here mean the two periods of LGM-MH and PI-L21. This kind of expression is proved to be inappropriate and have been revised in whole manuscript. We now believe this paragraph is uneccessary and should be removed.

12. Line 151 and Figure 2: Please justify the comparison between warming from the LGM to MH versus PI to future warming. This should be done in the methods and then discussed in the discussion. There are very important mechanistic differences between mid-Holocene and future warming. Mid-Holocene warming was driven by changes in primarily summertime insolation whereas future warming is driven by greenhouse forcing. Some impacts are comparable, but differences in forcing mechanisms need addressed. It appears that you try to do some of this in the discussion, but it needs to be developed/clarified.

Yes, we agree that the rationality of comparison between different timescales is the basis of our study. We will address this in more details in the revised version.

13. Figure 2 and 3 captions. Please indicate why the maps have extreme missing data coverage. The significant regions are shown by the gridding…CRU has data over the regions which show no data…

Thanks for reminding me. The maps only show changes in global closed basins, and we have clarified it in captions.

14. Line 193: Please point the reader to the correlations (Table 4) before discussing them.

Done.

15. Line 241: Please point toward something to justify the statement that winter precipitation will play a dominant roll in future hydroclimate changes.

It is based on results from monthly precipitation changes of L21-E21 (Table 2).

16. Line 256: Please remove the word "comprehensive".

 Done.

17. Line 266: The conclusion that moisture changes in closed basins are resilient to warming needs justified…Large increases in temperature alone will dramatically increase evaporation and decrease effective moisture (e.g. lake level) under RCP 8.5 scenarios.

It's more resilient than previous thought, especially compared to alarming declines in water storage in recent years. We will expand the discussion on this in the revised version.

18. Data availability: Please make your compilation of 52 hydroclimate records available in addition to data that were already available.

OK, a new statement was added.

19. In general, make sure statements are accompanied by the data that support them (Figures, tables or otherwise).

We had an overall check and revised.

---

## Author Comment (AC2) · 2 May 2020

**Response to Anonymous Referee #2**

**We very much thank the reviewer for taking the time to review our manuscript. The comments and suggestions are of tremendous assistance to improve the quality of our manuscript. The point-by-point responses are listed below.**

Dear editor and authors of the manuscript "Wet/dry status change in global closed basins between the mid-Holocene and the Last Glacial Maximum and its implication for future projection", I have no ability to assess the use of lake sediment data due to my professional restriction, and propose my opinion about the model results. This work is encouraged to help future projection using the paleoclimate information. The Global model-data comparison over the orbital scale and short-scale is important to understand the difference in the local hydroclimate variations. The analysis is logical and fruitful. I was attracted by the idea of the manuscript, but still felt unsatisfied about the mechanism. Thus, I suggest that the manuscript should be accepted for publication after a minor revision.

Main comments:

1. The mechanism should be furtherly improved. e.g. the influence of the insolation is not derived from the current study. A regional difference and its reason should be emphasized in the abstract and the conclusion. e.g. the difference of the hydroclimate variation mechanisms in the Central Eurasia and other regions.

To address this, we will rewrite the abstract and conclusion and discuss more details in the revised version,

2. The reason for choosing the analysis period is obscure. It is hard to understand to compare a centennial and longer variation of glacial-interglacial cycle with a decadal variation (AD 2006-2015 and AD 2091-2100). Is it better to choose the period as long as possible, e.g. the entire 20th century and 21th century?

We agree that the rationality of comparison between different timescales is the basis of our study. Even the entire 20th century and 21th century still mismatch the MH variation. Here we just consider the E21 and L21 as specific climate status. We will address this in more details in the revised version.

3. The conclusion should be furtherly verified. It is challenging that a decadal variation in the late 21th century is attributed to the ENSO variability. The Pacific decadal oscillation or the Inter-decadal pacific oscillation may be more appropriate, if the analyzed period would be extended to the entire century.

Thanks for your suggestion. A new table was added in the supplement, containing pearson correlation coefficients between AI and monthly NAO, SOI, PDO and TPI. The PDO indeed show significant correlation, but not better than ENSO does. We will address this in more details in the revised version.

Specific Comments:

1. Page 1, line 17. The location of these basins should be provided, which is your contribution. e.g. There is an opposite significant AI-MEI relationship between in the Southern Africa and in the Central Eurasia.

Added as suggested.

2. Page 2, line 30. The abbreviation of the term 'wet get wetter dry get drier' could be revised to 'DGDWGW' according to the previous study [Hu et al., 2019].

Revised and cited.

3. Page 2, line 56. 'Zhan et al.,'

Done.

4. Page 3, line 73-75. How to know that the proxy should be indicative of moisture changes and its drive is climatic change'? Following the original descriptions?

Yes, only records verified in the original study were used.

5. Page 3, line 83. What does the 'direction' mean? Is it a trend?

It means the positive or negative change of simulated effective precipitation from the LGM to MH. The sentence was rewritten to clarify it.

6. Pages 3-5. The table 1 should be changed with a Figure 1b to show model-data comparison.

Done.

7. Page 6, line 120. References of these experiments should be added.

Done.

8. Page 7, line 138-143. A new Figure 1b would be helpful to describe this part.

A new Figure 1b was added.

9. Page 7, line 153. The L21-PI difference is a future scenario not a modern warming.

Expression of modern warming was removed.

10. Page 8, line 174. Why to select the period 1979-2016? Is it better to choose the same period with the early 21th century (AD 2006-2015)?

To match the MEI data series (new version). Another reason is that historical observations used for CRU gridding are sparse or simply unavailable over many land areas during the first half of the 20th Century. The early 21th century (AD 2006-2015) is too short to capture the modern moisture trends.

11. Page 10, line 205. 'within'

Done.

12. Page 11, line 239. '2091-2100'?

Done.

13. Page 11, line 242. the period (AD 1901-2100) was not analyzed in this study.

Removed.

14. Page 11, lines 244-254. If the period is extended to the entire 20th century and 21th century, this discussion may be related to the above mechanism about a centennial and longer variation of glacial-interglacial cycle.

Thanks for your suggestion. We will expand this part of discussion in the invised version.

[Figure]

**Figure 1.** Wet/dry status change between the LGM and MH from lake records (a) and comparison with the simulated effective precipitation from PMIP3/CMIP5 multi-models (b).

**Table S2.** Pearson correlation coefficients between AI and monthly NAO, SOI, PDO and TPI during 1979-2016. The bold numbers mean that correlation coefficients are statistically significant at 5% level. SAM-South America, NAM-North America, SAF-Southern Africa, EAF-Eastern Africa, NAF-Northern Africa and Arabian peninsula, CEA-Central Eurasia, AUS-Australia, ALL-Global closed basins.

| | | 1 | 2 | 3 | 4 | 5 | 6 | 7 | 8 | 9 | 10 | 11 | 12 |
|---|---|---|---|---|---|---|---|---|---|---|---|---|---|
| **NAO** | ALL | 0.25 | -0.03 | 0.15 | -0.03 | -0.02 | -0.02 | -0.09 | 0.10 | -0.03 | 0.19 | 0.06 | 0.12 |
| | SAM | -0.15 | 0.00 | 0.05 | -0.02 | 0.11 | 0.00 | 0.02 | -0.07 | **-0.34** | 0.08 | 0.14 | 0.10 |
| | NAM | 0.01 | -0.08 | -0.11 | 0.10 | 0.11 | 0.07 | 0.09 | 0.10 | -0.17 | 0.18 | 0.18 | 0.45 |
| | SAF | -0.09 | -0.17 | 0.08 | 0.17 | 0.13 | **-0.36** | -0.11 | -0.25 | 0.01 | -0.18 | -0.06 | -0.07 |
| | EAF | 0.19 | 0.12 | **0.41** | -0.04 | -0.10 | -0.06 | -0.07 | 0.11 | 0.30 | 0.29 | -0.03 | 0.03 |
| | NAF | 0.06 | -0.05 | 0.15 | 0.05 | -0.04 | -0.12 | 0.06 | -0.08 | 0.18 | -0.11 | -0.02 | 0.04 |
| | CEA | **0.37** | -0.05 | -0.08 | -0.16 | -0.04 | 0.14 | -0.04 | 0.18 | -0.12 | 0.05 | 0.01 | 0.02 |
| | AUS | -0.17 | 0.04 | -0.06 | 0.26 | -0.08 | -0.20 | -0.03 | -0.08 | -0.13 | -0.03 | 0.09 | 0.13 |
| **SOI** | ALL | **-0.35** | **-0.35** | **-0.42** | **-0.48** | -0.07 | 0.16 | 0.01 | 0.07 | -0.03 | 0.21 | 0.12 | -0.08 |
| | SAM | -0.17 | 0.02 | -0.10 | -0.24 | -0.32 | -0.22 | -0.27 | -0.06 | -0.16 | -0.15 | -0.05 | -0.25 |
| | NAM | -0.01 | -0.09 | -0.06 | -0.03 | -0.17 | -0.19 | -0.24 | -0.13 | -0.12 | -0.15 | -0.07 | 0.03 |
| | SAF | -0.06 | -0.01 | -0.15 | -0.01 | -0.07 | 0.08 | -0.01 | -0.05 | 0.07 | 0.17 | 0.14 | 0.10 |
| | EAF | -0.22 | **-0.34** | **-0.35** | -0.15 | 0.16 | 0.21 | 0.14 | 0.03 | 0.01 | 0.22 | -0.01 | 0.08 |
| | NAF | -0.17 | -0.13 | 0.07 | -0.15 | 0.22 | 0.07 | 0.02 | -0.02 | 0.18 | 0.15 | 0.05 | 0.09 |
| | CEA | -0.26 | -0.30 | **-0.37** | **-0.53** | -0.23 | 0.01 | -0.08 | 0.03 | -0.14 | -0.01 | -0.01 | -0.23 |
| | AUS | 0.19 | **0.39** | 0.27 | 0.13 | 0.07 | -0.14 | -0.08 | 0.01 | 0.09 | 0.11 | 0.21 | 0.04 |
| **PDO** | ALL | -0.10 | -0.06 | -0.07 | 0.06 | 0.08 | 0.16 | 0.19 | 0.29 | 0.31 | 0.30 | **0.33** | **0.34** |
| | SAM | -0.19 | -0.17 | -0.13 | -0.12 | -0.07 | -0.04 | 0.01 | 0.12 | 0.15 | 0.15 | 0.14 | 0.06 |
| | NAM | **0.51** | **0.57** | **0.56** | **0.48** | **0.39** | 0.29 | 0.19 | 0.13 | 0.08 | 0.06 | 0.05 | 0.07 |
| | SAF | **-0.60** | **-0.63** | **-0.61** | **-0.56** | **-0.49** | **-0.43** | **-0.33** | -0.22 | -0.13 | -0.07 | 0.00 | 0.03 |
| | EAF | -0.09 | -0.09 | -0.11 | -0.03 | 0.01 | 0.08 | 0.05 | 0.06 | 0.13 | 0.14 | 0.19 | 0.23 |
| | NAF | 0.15 | 0.10 | 0.07 | 0.11 | 0.05 | 0.00 | -0.03 | -0.02 | 0.02 | 0.07 | 0.13 | 0.10 |
| | CEA | 0.08 | 0.14 | 0.14 | 0.27 | 0.29 | **0.37** | **0.42** | **0.48** | **0.41** | **0.37** | **0.34** | **0.37** |
| | AUS | 0.06 | 0.09 | 0.08 | 0.00 | -0.09 | -0.19 | -0.24 | -0.26 | -0.28 | **-0.32** | **-0.34** | **-0.34** |
| **TPI** | ALL | -0.10 | -0.06 | -0.07 | 0.06 | 0.08 | 0.16 | 0.19 | 0.29 | 0.31 | 0.30 | **0.33** | **0.34** |
| | SAM | -0.19 | -0.17 | -0.13 | -0.12 | -0.07 | -0.04 | 0.01 | 0.12 | 0.15 | 0.15 | 0.14 | 0.06 |
| | NAM | **0.51** | **0.57** | **0.56** | **0.48** | **0.39** | 0.29 | 0.19 | 0.13 | 0.08 | 0.06 | 0.05 | 0.07 |
| | SAF | **-0.60** | **-0.63** | **-0.61** | **-0.56** | **-0.49** | **-0.43** | **-0.33** | -0.22 | -0.13 | -0.07 | 0.00 | 0.03 |
| | EAF | -0.09 | -0.09 | -0.11 | -0.03 | 0.01 | 0.08 | 0.05 | 0.06 | 0.13 | 0.14 | 0.19 | 0.23 |
| | NAF | 0.15 | 0.10 | 0.07 | 0.11 | 0.05 | 0.00 | -0.03 | -0.02 | 0.02 | 0.07 | 0.13 | 0.10 |
| | CEA | 0.08 | 0.14 | 0.14 | 0.27 | 0.29 | **0.37** | **0.42** | **0.48** | **0.41** | **0.37** | **0.34** | **0.37** |
| | AUS | 0.06 | 0.09 | 0.08 | 0.00 | -0.09 | -0.19 | -0.24 | -0.26 | -0.28 | **-0.32** | **-0.34** | **-0.34** |

---

## Author Response (AR1)

Dear editor and reviewers,

We would like to thank you for your kind letter and for reviewers' constructive comments concerning our manuscript entitled "Wet/dry status change in global closed basins between the mid-Holocene and the Last Glacial Maximum and its implication for future projection" (Manuscript No.: cp-2020-21).These comments are all valuable and helpful for improving our article. According to the reviewers' comments, we have tried best to modify our manuscript to meet with the requirements of the journal. In this revised version, changes to our manuscript within the document were all highlighted by using red colored text. Point-by-point responses to the reviewers are listed below this letter.

Yours sincerely,
Xinzhong Zhang

Corresponding author:
Name: Yu Li
Address: College of Earth and Environmental Sciences, Lanzhou University, Lanzhou 730000, China
E-mail: liyu@lzu.edu.cn

**Response to Referee #1**

**General response: We very much appreciate the careful reading of our manuscript and valuable suggestions of the reviewer. Referee #1 focus on more clarified writing. According to the suggestions, we totally rewritten the results and extensively revised the discussion, abstract and conclusion. A new Figure 1b was added and other changes were marked by red colored text in the revised version. The point-by-point responses are listed below.**

Zhang et al., present a nice new compilation of existing Holocene and glacial hydroclimate records, which is accompanied by an interesting analysis. They compare proxy hydroclimate records, proxy lake level records, and PMIP simulations between the last glacial maximum, mid-Holocene, and future warming. Generally the analyses are straight forward and I think worthwhile of publication. However, before I can adequately evaluate the study design and associated conclusions, much work needs to be done to clarify the writing. The manuscript would substantially improve by having anative english speaker edit the sentences and overall structure. Many of the sentences are incomplete, difficult to follow, or entirely nonsense. I would also highly recommend reading and implementing the writing principals outlined by Joshua Schimel in his book "Writing Science" published in 2012. Below is my preliminary review, which was conducted rapidly and remains incomplete until the overall presentation and writing is improved.

1. The introduction is meandering and hard to follow. Make sure each sentence doing work, frame the knowledge gap, and keep the story moving forward in a logical sequence.

Thanks for your suggestion. We have revised the introduction to improve readability.

2. Line 62. This gets to the point of the study, but should also include the LGM. Something along the lines of "…pattern of changes during the LGM, mid-Holocene and modern warm period…" Also there is only one modern warm period so it should be singular in this sentence.

The sentence was rewritten as suggested.

3. Line 71: This sentence is difficult for me to follow. Please re-write. Or remove?

Thanks for your suggestion. This sentence was removed.

4. Line 72: Is there a sampling resolution criteria?

The dating control level (descripted in next answer) rather than a sampling resolution criteria was used here.

5. Line 76: I appreciate a description of the COHMAP dating scheme, however it is difficult to follow as written. Please clarify. Use multiple sentences if needed.

The description was rewritten as follows: The records should have a dating control level of 6 or better for 21 ka and 6 ka time slices according to the Cooperative Holocene Mapping (COHMAP) project dating scheme. A dating control level of 6 for continuous sequences was based on the following criteria: Bracketting dates, one within 6000 years and the other within 8000 years or one within 4000 years and the other within 10000 years of the selected date (21 ka and 6 ka). The same control level applied to discontinuous sequences requires at least one date within 2000 years of the time being assessed (Street-Perrott et al., 1989; COHMAP Members, 1994; Lowry & Morrill, 2019).

6. Line 79. It took me some time to figure out what you are trying to communicate here. The finding of 52 sites and Table 1 are results, and should be moved down into the beginning of the results section. Then, on line 79, a new paragraph should be started with rewording the sentence to something along the lines of: "We then compared our new compilation of proxy records to 50…"

Thanks for your suggestion. A new Figure 1b (model-data comparison based on Table 1) was added, and Table 1 was moved into Supplement. On original line 79, a new paragraph was started with rewording the sentence as suggested.

7. Line 82-84: Sentence structure, please clarify. Use multiple sentences as necessary.

The sentence was clarified using multiple sentences as follows: To capture the general spatial pattern, the differences of lake status between the LGM and MH in individual records were classified into 3 grades (higher/wetter, moderate, lower/drier). Similarly, the differences of simulated effective precipitation between the LGM and MH from PMIP3/CMIP5 multi-models in certain grid of the lake site were classified into positive, no change and negative and compared with the records.

8. Line 107: Replace "involved" with "used"

Thanks for your suggestion. Revised.

9. Line 127: This sentence doesn't make sense.

The whole paragraph was reorganized and revised to make it clearer.

10. Line 138: Please show the data (a graph or otherwise) to support the statement in this paragraph.

Thanks for your suggestion. A new Figure 1b was added.

11. Line 145: I'm not following the argument in this paragraph. What two global warming processes? Was this described somewhere in the methods? E21 and L21? Please clarify.

Two global warming processes here mean the two periods of LGM-MH and PI-L21. We realized this kind of expression is ambiguous and inappropriate and has been removed in the manuscript.

12. Line 151 and Figure 2: Please justify the comparison between warming from the LGM to MH versus PI to future warming. This should be done in the methods and then discussed in the discussion. There are very important mechanistic differences between mid-Holocene and future warming. Mid-Holocene warming was driven by changes in primarily summertime insolation whereas future warming is driven by greenhouse forcing. Some impacts are comparable, but differences in forcing mechanisms need addressed. It appears that you try to do some of this in the discussion, but it needs to be developed/clarified.

Thanks for your suggestion. Indeed, the two periods are at very different timescales. Here, we focus on the comparable hydroclimate responses to them and what we can learn from them rather than the comparison itself. As mentioned in the revised discussion, "Thus, their climate changes are strongly influenced by the interactions of mid-latitude westerlies and low-latitude monsoon especially on a long-term timescale. Given the position of westerlies and strength of monsoons in modern times have no chance to change dramatically as in the last deglaciation, the oceans rather than the Earth orbital forcing start to play more important roles in controlling the regional moisture change. Even so, hydroclimate change in some closed basins respond in the same pattern to the past and future warming, indicating deeper connections between different timescales. More importantly, the long-term hydroclimate change patterns provide the baseline for modern and future climate change assessment.".

13. Figure 2 and 3 captions. Please indicate why the maps have extreme missing data coverage. The significant regions are shown by the gridding…CRU has data over the regions which show no data…

Thanks for your reminding. The maps only show changes in global closed basins, and we have clarified it in captions.

14. Line 193: Please point the reader to the correlations (Table 4) before discussing them.

Thanks for your suggestion. Revised.

15. Line 241: Please point toward something to justify the statement that winter precipitation will play a dominant roll in future hydroclimate changes.

It is based on results from monthly precipitation changes of L21-E21 (Table 2) and we have clarified it in the new statement.

16. Line 256: Please remove the word "comprehensive".

Thanks for your suggestion. Revised.

17. Line 266: The conclusion that moisture changes in closed basins are resilient to warming needs justified…Large increases in temperature alone will dramatically increase evaporation and decrease effective moisture (e.g. lake level) under RCP 8.5 scenarios.

Thanks for your suggestion. Here we mean it's more resilient than previous thought, especially compared to alarming declines in water storage in recent years. By reconsidering it, we believe this conclusion was less rigorous and redundant. Thus, we have revised the abstract and conclusion after removing it.

18. Data availability: Please make your compilation of 52 hydroclimate records available in addition to data that were already available.

Thanks for your suggestion. A new statement was added.

19. In general, make sure statements are accompanied by the data that support them (Figures, tables or otherwise).

Thanks for your suggestion. We have checked and revised the whole manuscript.

**Response to Anonymous Referee #2**

**General response: We very much appreciate the careful reading of our manuscript and valuable suggestions of the reviewer. Anonymous Referee #2 focus on furtherly improved mechanism and furtherly verified conclusion. According to the suggestions, we extensively revised the discussion, abstract and conclusion. A table resulting from new data analysis was added and other changes were marked by red colored text in the revised version. The point-by-point responses are listed below.**

Dear editor and authors of the manuscript "Wet/dry status change in global closed basins between the mid-Holocene and the Last Glacial Maximum and its implication for future projection", I have no ability to assess the use of lake sediment data due to my professional restriction, and propose my opinion about the model results. This work is encouraged to help future projection using the paleoclimate information. The Global model-data comparison over the orbital scale and short-scale is important to understand the difference in the local hydroclimate variations. The analysis is logical and fruitful. I was attracted by the idea of the manuscript, but still felt unsatisfied about the mechanism. Thus, I suggest that the manuscript should be accepted for publication after a minor revision.

Main comments:

1. The mechanism should be furtherly improved. e.g. the influence of the insolation is not derived from the current study. A regional difference and its reason should be emphasized in the abstract and the conclusion. e.g. the difference of the hydroclimate variatiocn mechanisms in the Central Eurasia and other regions.

Thanks for your suggestion. To further improve the mechanism, we have reorganized and revised the discussion in a more reasonable way. And according to the suggestion, we have rewritten the abstract and conclusion in the revised version.

2. The reason for choosing the analysis period is obscure. It is hard to understand to compare a centennial and longer variation of glacial-interglacial cycle with a decadal variation (AD 2006-2015 and AD 2091-2100). Is it better to choose the period as long as possible, e.g. the entire 20th century and 21th century?

Thanks for your suggestion. In this study, we aim to verify the hydroclimate responses to the different warm periods rather than to compare different warm periods at the same timesclae. Even the entire 20th century and 21th century won't match the millennial variation from the LGM to MH. In fact, the period from the PI to late 21th century is the longest we can extend under modern and near-future warming, and can emphasize the hydroclimate responses more obviously. That's why we use this period comparing with MH-LGM. Here we just consider the E21 and L21 as specific climate status like the PI condition.

3. The conclusion should be furtherly verified. It is challenging that a decadal variation in the late 21th century is attributed to the ENSO variability. The Pacific decadal oscillation or the Inter-decadal pacific oscillation may be more appropriate, if the analyzed period would be extended to the entire century.

Thanks for your suggestion. A new table was added in the supplement, containing pearson correlation coefficients between AI and monthly NAO, SOI, PDO and TPI. As results, the performance of NAO, SOI and PDO are comparatively weak, and the MEI responds the best and shows the similar pattern as TPI does, both indicating the dominant role of the Pacific Ocean oscillation in controlling the moisture change of global closed basins (Table 3, Supplement Table S2).

Specific Comments:

1. Page 1, line 17. The location of these basins should be provided, which is your contribution. e.g. There is an opposite significant AI-MEI relationship between in the Southern Africa and in the Central Eurasia.

Thanks for your suggestion. A new statement was added as follows: However, modern moisture changes show correlations with ENSO in most closed basins, namely the opposite significant AI-MEI relationships between the North America and the Southern Africa and between the Central Eurasia and the Australia, indicating strong connection with ocean oscillation.

2. Page 2, line 30. The abbreviation of the term 'wet get wetter dry get drier' could be revised to 'DGDWGW' according to the previous study [Hu et al., 2019].

Thanks for your reminding. The abbreviation was revised and cited.

3. Page 2, line 56. 'Zhan et al.,'

Sorry for our carelessness. Revised.

4. Page 3, line 73-75. How to know that the proxy should be indicative of moisture changes and its drive is climatic change'? Following the original descriptions?

Yes, only records verified in the original study were used.

5. Page 3, line 83. What does the 'direction' mean? Is it a trend?

Thanks for your reminding. It means the positive or negative change of simulated effective precipitation from the LGM to MH. The sentence was rewritten to clarify it as follows: To capture the general spatial pattern, the differences of lake status between the LGM and MH in individual records were classified into 3 grades (higher/wetter, moderate, lower/drier). Similarly, the differences of simulated effective precipitation between the LGM and MH from PMIP3/CMIP5 multi-models in certain grid of the lake site were classified into positive, no change and negative and compared with the records.

6. Pages 3-5. The table 1 should be changed with a Figure 1b to show model-data comparison.

Thanks for your suggestion. Done.

7. Page 6, line 120. References of these experiments should be added.

Thanks for your suggestion. Done.

8. Page 7, line 138-143. A new Figure 1b would be helpful to describe this part.

Thanks for your suggestion. A new Figure 1b was added.

9. Page 7, line 153. The L21-PI difference is a future scenario not a modern warming.

Thanks for your reminding. Expression of modern warming was removed in 3.2 section.

10. Page 8, line 174. Why to select the period 1979-2016? Is it better to choose the same period with the early 21th century (AD 2006-2015)?

This was to match the MEI data series (new version). Another reason is that historical observations used for CRU gridding are sparse or simply unavailable over many land areas during the first half of the 20th Century. And the early 21th century (AD 2006-2015) is too short to capture the modern moisture trends.

11. Page 10, line 205. 'within'

Sorry for our carelessness. Revised.

12. Page 11, line 239. '2091-2100'?

Sorry for our carelessness. Revised.

13. Page 11, line 242. the period (AD 1901-2100) was not analyzed in this study.

Sorry for that mistake. This part was removed.

14. Page 11, lines 244-254. If the period is extended to the entire 20th century and 21th century, this discussion may be related to the above mechanism about a centennial and longer variation of glacial-interglacial cycle.

Thanks for your comment. The reasons why we didn't extend the period to the entire 20th century and 21th century have been mentioned in the anwser for main comment 2 and specific comment 10. And we have revised whole discussion to improve it.

---

## Author Response (AR2)

Dear editor and reviewers,

Many thanks for your kind letter and for reviewers' constructive comments concerning our manuscript entitled "Wet/dry status change in global closed basins between the mid-Holocene and the Last Glacial Maximum and its implication for future projection" (Manuscript No.: cp-2020-21). According to the reviewers' comments, we have revised our manuscript to improve the statement and discussion and to meet with the requirements of the journal. In this revised version, changes to our manuscript within the document were all highlighted by using red colored text. Point-by-point responses are listed below.

Yours sincerely,
Xinzhong Zhang

Corresponding author:
Name: Yu Li
Address: College of Earth and Environmental Sciences, Lanzhou University, Lanzhou 730000, China
E-mail: liyu@lzu.edu.cn

**General response: We very much appreciate the careful reading of our manuscript and valuable suggestions of the reviewer. Referee suggested the mechanisms of change need developed through a more thorough review of the literature. Accordingly, we have corrected the inappropriate statements and extensively revised the discussion. A new Figure 4 was added in the discussion section and other changes were marked by red colored text in the revised version. The point-by-point responses are listed below.**

Zhang et al., have presented a substantially improved manuscript. I appreciate their efforts to dive in and clarify the writing and presentation. Overall I think the manuscript is an important contribution. My main concern is that the mechanisms of change need development and support. The authors present compelling patterns of change (LGM to Mid Holocene and Preindustrial vs 21st century). But the mechanisms of change need work. The authors state in the abstract: "The long-term pattern of moisture change is highly related to the high-latitude ice sheets and low-latitude solar radiation, which leads to the poleward moving of westerlies and strengthening of monsoons during the interglacial period." I agree with this statement, but these concepts are poorly developed in the manuscript. As presented, there are some poorly referenced conceptual ideas in the discussion, but a limited amount of text to support the above statement. At minimum the mechanisms of change need developed through a more thorough review of the literature. Bhattacharya et al. (2018); Routson et al. (2019); and Ramisch et al., (2016) would be some good papers to start with. Ideally the authors would present some evidence that their proposed mechanisms were indeed the cause of the observed changes, rather than simply associations.

A few line by line comments follow:

1. Line 11: Delete 'the' in this sentence: "By integrating lake records, …

Thanks for your suggestion. We have revised it and checked other parts.

2. Lines 16 and 17: Please define abbreviations before use (ENSO AI-MEI).

Thanks for your suggestion. To avoid redundancy in the Abstract, we have clarified the sentence as follows "However, modern moisture changes show correlations with El Niño/Southern Oscillation in most closed basins, such as the opposite significant correlations between North America and Southern Africa and between Central Eurasia and Australia, indicating strong connection with ocean oscillation."

3. Line 101: A significance level of 95%…?

We have checked it in other literatures and the original statement is indeed right.

4. Line 127: Please site a reference that suggests hydroclimate records from western China are controversial. Or remove statement.

Thanks for your suggestion. we have removed the statement and revised the sentence as this "Generally, lake level changes match climate changes from the proxy records well except for Central Asia (Fig. 1)."

5. Table 2: Label numbers as months.

Thanks for your suggestion. We have changed it in all tables.

6. Line 164: October is wetter than July. So the wettest months = June-November, or July - October rather than July - September.

Thanks for your suggestion. Here we mean that the average increasing of precipitation during July-September is wettest. To avoid ambiguity, we have revised it accordingly "The precipitation increases about 50% from the LGM to the MH in the wettest months of July-October, 13% from PI to L21 in the wettest months of February-April."

Line 165: This is an odd result that evaporation diminishes in the future when there will be strong warming. Am I reading this correctly?

Thanks for your reminding. Here we refer to the seasonal difference in evaporation and precipitation. We have clarified it as "Seasonal variation in evaporation is smaller than that in precipitation but keep the same pattern. In addition, precipitation and evaporation changes from E21 to L21 make significant contributions to the increasing of precipitation and evaporation from PI to L21, especially in the boreal summer half-year."

Line 205: This sentence is confusing. I think I'm tracking now, but please clarify.

Thanks for your suggestion. We have revised the discussion to improve the readability and clarify this sentence in other part as follows "On a shorter timescale of modern times, strengthening or moving of monsoons and westerlies are largely limited compared to that from the LGM to MH."

References:

Bhattacharya, T., Tierney, J. E., Addison, J. A. and Murray, J. W.: Ice-sheet modulation of deglacial North American monsoon intensification, Nature Geoscience, 11(11), 848–852, doi:10.1038/s41561-018-0220-7, 2018.

Ramisch, A., Lockot, G., Haberzettl, T., Hartmann, K., Kuhn, G., Lehmkuhl, F., Schimpf, S., Schulte, P., Stauch, G., Wang, R., Wünnemann, B., Yan, D., Zhang, Y. and Diekmann, B.: A persistent northern boundary of Indian Summer Monsoon precipitation over Central Asia during the Holocene, Science Reports, 6.

Routson, C. C., McKay, N. P., Kaufman, D. S., Erb, M. P., Goosse, H., Shuman, B. N., Rodysill, J. R. and Ault, T.: Mid-latitude net precipitation decreased with Arctic warming during the Holocene, Nature, 568(7750), 83–87, doi:10.1038/s41586-019-1060-3, 2019.

---

## Author Response (AR3)

Dear Editor,

Thanks very much for your kind work and consideration on publication of our paper. We really appreciate it and do learned a lot in the process. On behalf of my co-authors, we would like to express our great appreciation to you and reviewers and wish you all the best in your future work. In this version, we revised the references format according to the guidelines and made some changes to the Acknowledgements, the marked-up manuscript is attached below.

Best regards,

Xinzhong Zhang

Corresponding author:

Name: Yu Li

Address: College of Earth and Environmental Sciences, Lanzhou University, Lanzhou 730000, China

E-mail: liyu@lzu.edu.cn

[revised manuscript text omitted]